# Students' Affective Learning Outcomes and Academic Performance in the Blended Environment at University: Comparative Study

**Aleksandra Kobicheva \*, Elena Tokareva and Tatiana Baranova**

Institute of Humanities, Peter the Great Saint-Petersburg Polytechnic University, 195251 St. Petersburg, Russia
\* Correspondence: kobicheva_am@spbstu.ru or kobicheva92@gmail.com

**Abstract:** This study examines how the gender of students and their level of education (undergraduate or postgraduate students) influence affective learning outcomes in a mixed environment. The research model is based on four key indicators: satisfaction, perception of experience (perceived usefulness, ease of use, and perceived behavioral control), perception of benefits (general learning effectiveness, knowledge sharing and increasing, study skills improvement, and sense of progress), and academic performance. Moderating factors, including gender and education level, were used to define the distinctions in the measured results. The study was conducted in the fall semester of 2021. The target samples were students of undergraduate and postgraduate levels studying during the semester in the blended environment. A total of 513 students from Peter the Great Polytechnic University took part in the research. The online questionnaire was conducted to define the affective learning outcomes of students in the blended environment and the influence of affective outcome factors on academic performance. The Likert-type five-point scale was used to determine all the variables. For our statistical analysis, we used SPSS 24.0 and SmartPLS 3.0 programs. Two tests were implemented to determine the differences between gender and education level in relation to students' affective learning outcomes. Finally, our study revealed how distinctive aspects of affective learning outcomes impact academic learning outcomes in a blended format using linear regression analysis. According to the results obtained, the results between males and females were similar and academic performance results were mostly predicted by satisfaction level. A difference was revealed between undergraduate students' results and postgraduate students' results. Perceived benefit has a greater effect on academic performance for postgraduate students, while satisfaction level has a greater effect on academic performance results for undergraduate students than for postgraduate students.

**Keywords:** affective learning outcomes; student's affective outcomes; academic performance; blended learning; e-learning

## 1. Introduction

Blended learning is a form of combining online learning with traditional learning. The introduction of blended learning has changed the way higher education resources are provided. Although blended learning was initially used as non-formal learning, during the pandemic, the issue of introducing blended learning as part of formal campus education was especially relevant. Blended learning has been implemented in various practices, from undergraduate to postgraduate education. Learning assessment is one of the most important aspects [1]. A variety of assessment methods naturally emerge in response to the need to measure learning outcomes. While there is debate about the various methods that can be used as an assessment of learning in a blended environment, they have not paid attention to the relationship between learning outcomes and assessment tools in a wide range of higher education disciplines. Evaluation should be the main factor at the very beginning of course design in a blended environment, not a later addition [2]. However, due to the COVID-19 pandemic, the situation has not given enough time to

develop quality assessment tools. The alignment between learning objectives, learning outcomes, and assessment tools is important to ensure that learners achieve the necessary course achievement. Assessing learning outcomes is key to measuring actual student achievement, which is also critical to examining the effectiveness of teaching methods and student learning [1].

This paper investigates university students' affective learning outcomes and academic performance in a blended environment in the academic years 2020–2021. This is a comparative study which has used gender and education level to define the differences in the measured indicators. Factors identifying affective learning outcomes in a blended format were analyzed.

Specifically, this study focused on three major research questions:

1. Are students' affective learning outcomes and academic performance in blended learning affected by gender?
2. Are students' affective learning outcomes and academic performance in blended learning affected by the education levels of students' degrees, including undergraduate (UG), and post-graduate (PG) degrees?
3. To what extent do students' affective learning outcomes predict students' academic performance? Is there a difference by gender or by education level?

The structure of the paper is as follows: Section 1 presents the theoretical background to the investigation; Section 2 describes the developed research model and hypothesis as well as the methodology of the current research; in Section 3, the validity testing results and analysis of students' affective learning outcomes and academic performance in a blended learning environment are discussed; Section 4 presents the comparison of our research with previous studies; and Section 5 concludes the findings.

### 1.1. Background

#### 1.1.1. Blended Learning

Online learning has become widespread due to the development of technology [1–4]. The consequence of the COVID-19 pandemic has been a change in the format of the educational process, including online or blended learning [5–7].

For the current study, blended learning refers to the integration of two types of learning—online learning and face-to-face learning [8–10]. This educational model has such features as flexibility, access to a variety of educational resources, and the interactivity of traditional learning.

Early research on blended learning focused on the theoretical aspects of the educational model, the use of resources and learning content, the creation of a supportive educational environment, and the evaluation of the effectiveness of blended learning in several dimensions, such as educational outcomes, skill development, and learning satisfaction [11–15]. Most research findings reflect the effectiveness of blended learning versus face-to-face learning [16–21]. There are also studies [22,23] showing a positive attitude towards blended learning through flexible learning opportunities and promoting innovation, which deserves further research.

#### 1.1.2. Affective Outcomes

Affective outcomes indicate students' perceptions of their education in a blended environment. To determine aspects of affective learning in a blended environment, student satisfaction with the course (positive assessment of courses), perceptions of the learning experience (students' perceptions of evaluation of learning on the course), and benefits (students' perception of improved learning) were examined.

Satisfaction was examined in seven studies. As a result of studies, participants in courses in a blended learning environment have given their courses positive ratings [24–26]. For example, de Lima and Zorrilla [27] assessed student satisfaction with the achievement of learning objectives. The results of the study indicated that the students were completely satisfied with the communication with their peers, especially highlighting the support

and comments from peers as well as the sharing of resources by other participants. K. Lee [28] and Rabin and Kalman and Kaltz [29] also evaluated student satisfaction with the educational process. A study by Watson, Watson, Yu, Alamri, and Muller [30] examined student satisfaction more specifically. The applicants of the course expressed their satisfaction with the study of the course, the learning process, and tasks. Comparative analysis revealed that men's satisfaction with the course was lower than that of women. In addition, instructor-led students were less satisfied than students doing research.

Studies on the emotional state of students were considered. For example, one study [31] explored the perceptions of students who report that the course is boring or, on the contrary, are satisfied with the course. Loizzo et al. [32] studied the concept of enjoyment students get from the educational process in a mixed environment. In a study by L. Wang, Hu, and Zhou [33], students' emotional tendencies (joy, sadness, anger, disappointment, surprise, pride, falling in love, and fear) were analyzed. We also reviewed studies examining participant perceptions of learning [25,34,35]. For example, Poquet et al. [36] and Kovanovich et al. [37] focused on aspects related to the social presence of students (i.e., group cohesion, open communication, and emotional expression) in a mixed environment.

Studies on student attitudes towards such factors as course design and content, interaction with the teacher, simplicity and ease of use of the course, its usefulness, and control were also considered [29,34,37–41]. Moreover, in these studies, the problems that students faced while studying in a blended learning environment were considered. According to Shapiro et al. [42] and Watson et al. [30], students experienced a heavy workload, a lack of discussion practice, a lack of prior knowledge, and a lack of time to master the material [40,43].

Studies were reviewed that reflected the benefits of learning noted by students. Some studies have looked at the motivation of course participants [26,32] and evaluated the effectiveness of blended learning environments [38]. For example, Shapiro et al. [42] identified motivations for students to study online. Such as enrichment of knowledge and improvement of skills, and the development of a future career is especially highlighted. DeBoer, Haney, Atiq, Smith, and Cox [44] conducted a study to examine student self-efficacy in a blended learning environment on neuroscience topics. At the end of this course, students showed a significant increase in self-efficacy as a result of the study. Some studies [32,45] investigated students' perceptions of behavioral learning (i.e., affective, behavioral, and cognitive learning) in a blended learning environment. In addition, several studies have looked at other benefits of blended learning environments, such as learning effectiveness [34,43], knowledge sharing [40,46], improved learning skills [47], sense of progress [34], confidence [25,48], and user persistence [39,49].

The analyzed studies describe the factors that make it possible to study the attitudes of students and affective learning outcomes. However, we believe that these factors can be combined in different ways depending on the conditions of the study, forming one research model. The use of this type of analysis in a mixed educational environment has not been previously used in the literature. In addition, an important condition for such analysis is the epidemiological situation in which a mixed educational environment was introduced abruptly without prior preparation.

We fill these gaps in the literature by providing an integrated conceptual framework for examining the influence of various factors on affective learning outcomes and academic performance of students in a blended learning environment, given the perceived benefits, perceived experience, and student satisfaction. We consider these factors as a prior attitude towards learning within a blended environment, which, in turn, influences students' intention to continue learning and improve their academic performance.

Innovation diffusion theory [50] provides a theoretical framework for current research. Diffusion theory states that the decision to innovate consists of five stages: knowledge, persuasion, decision, implementation, and confirmation [50]. Based on Rogers' conceptualization, we further suggest seven proposed characteristics of innovation that influence

people's decisions to innovate: perceived usefulness, ease of use, perceived behavioral control, general learning effectiveness, knowledge sharing and increasing, study skills improvement, and sense of progress. We singled out these categories as they directly affect academic performance and students' attitudes towards a mixed environment. In the context of affective learning outcomes in a blended environment, which represents an innovation in the educational environment, this theory can help explain how the perception of innovative characteristics of blended learning affects affective learning outcomes.

## 2. Materials and Methods

### 2.1. Research Model

The training was built on flipped classroom activities and project-based learning as pedagogical tools to create a blended learning environment. Work on each module of the discipline is divided into four stages. The first stage of work consists of the independent work of students with theoretical material. For this, the Moodle electronic educational platform is used, which provides the necessary materials and links to additional sources of information.

At the classroom lesson (stage 2), students are given time to discuss the studied material, answering students' questions. Furthermore, various tasks for the application of theoretical material are proposed. During the lesson, students discuss new terminology that is incomprehensible in self-study and consider ways to apply the information in practice.

The basis of the 3rd stage is teamwork. It is necessary to divide the group into 3–4 small subgroups of students, each of which studies the proposed section of the theoretical material. Then, during a class session, students discuss the material they have learned with other groups, sharing key concepts and terms. At the end of each block of theoretical material, questions for reflection and analysis are presented. The whole group is invited to collectively answer the questions posed (teamwork). The electronic educational platform Moodle presents video and audio materials on the topics being studied, which allow you to better understand the theoretical material.

The final 4th stage consists of design work. Students are offered a task in the format of a case study with questions. After studying the case, students prepare a presentation, either in their teams or individually, about their proposed solution to the problem and answer questions (project class).

The research model (Figure 1) is based on four key indicators: satisfaction, perception of experience (perceived usefulness, ease of use, and perceived behavioral control), perception of benefits (general learning effectiveness, knowledge sharing and increasing, study skills improvement, and sense of progress), and academic performance. Moderating factors, including gender and education level, were used to define the differences in the measured outcomes.

In order to predict students' differences in affective outcomes and academic performance between genders and education level, the following hypotheses are put forward:

**Hypothesis 1a (H1a).** *Students' affective learning outcomes in blended environment are affected by gender.*

**Hypothesis 1b (H1b).** *Students' academic performance in blended environment is affected by gender.*

**Hypothesis 2a (H2a).** *Students' affective learning outcomes in blended environment are affected by education level.*

**Hypothesis 2b (H2b).** *Students' academic performance in blended environment is affected by education level.*

**Hypothesis 3 (H3).** *Students' affective learning outcomes significantly and positively affect academic performance in blended environment.*

**Hypothesis 4 (H4).** *Gender significantly influences the relationship between affective learning outcomes and academic performance.*

**Hypothesis 5 (H5).** *Education level significantly influences the relationship between affective learning outcomes and academic performance.*

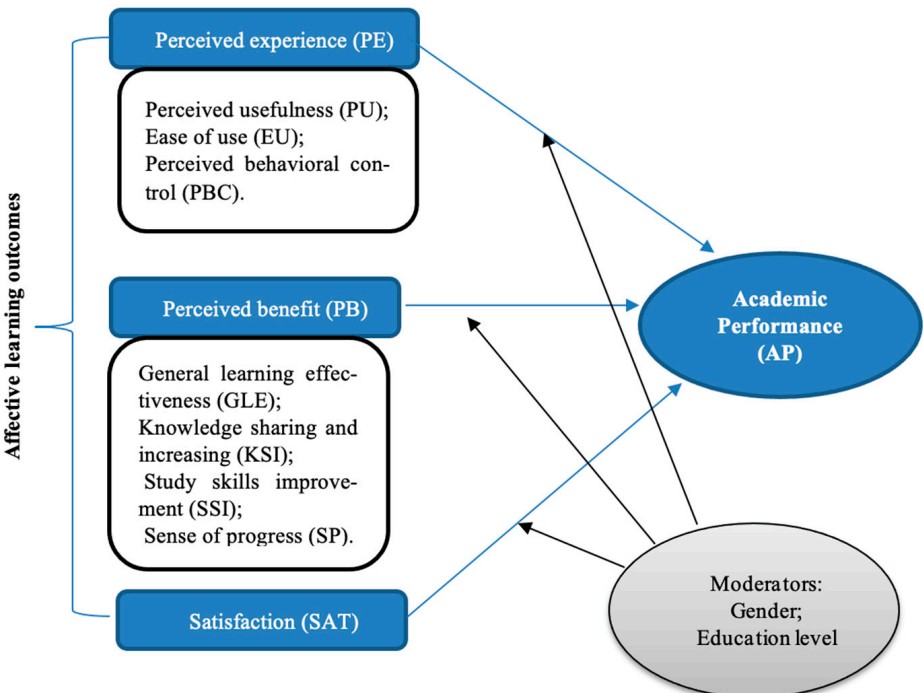

**Figure 1.** Research model.

*2.2. Data Collection*

The study was conducted in the fall semester of 2021 (15 learning weeks). The target samples were students of undergraduate and postgraduate levels studying during the semester in the blended environment. A total of 544 students of Peter the Great Polytechnic University took part in the online questionnaire (Appendix A) on a voluntary basis, 513 data were used in the study due to the incomplete nature of 31. According to the demographic data, 46.98% and 53.02% were male and female students, so gender distribution was quite balanced. All students were from humanities fields of study: 34.70%—students of linguistic sciences, 45.61%—students of legal sciences, and 19.69%—students of psychological sciences (Table 1). The age of the respondents ranged from 20 to 24 (mean age—22.2).

**Table 1.** Demographic data (N = 513).

| Demographic Variables | | Number | Percentage |
|---|---|---|---|
| Gender | Male | 241 | 46.98 |
| | Female | 272 | 53.02 |
| Education | Undergraduate | 403 | 78.56 |
| | Postgraduate | 110 | 21.44 |
| Field of study | Linguistics | 178 | 34.70 |
| | Psychology | 101 | 19.69 |
| | Law | 234 | 45.61 |

To measure academic performance, the results of professional disciplines were taken. Among postgraduate students, the academic results of the 1st year master students were considered. The online questionnaire was conducted in the Moodle system to define the

affective learning outcomes of students in a blended environment during the COVID-19 pandemic and the influence of affective outcome factors on academic performance. The Likert-type five-point scale was used to measure all the variables. Each question item was developed based on the existing literature and instruments in previous studies. The basis for the items measuring "perceived usefulness", "ease of use", and "perceived behavioral control" were the scales from Bhattacherjee [51], Ifinedo [52], and Yeap et al. [53]. To evaluate general learning effectiveness; knowledge sharing and increasing; study skills improvement; and sense of progress, we used eight items, each adopted from Valdivia Vázquez et al. [26] and Y. Jung & Lee [38]. We used the students' semester grades in professional disciplines (four courses) as measures of academic performance (AP). Official statements were used to collect information on grades.

### 2.3. Data Analysis

For our statistical analysis, we used SPSS 24.0 (version 24.0, IBM, Armonk, NY, USA) and SmartPLS 3.0 (version 3.0, SmartPLS GmbH, Oststeinbek, Germany) programs. To compare students' affective learning outcomes scores in the blended environment, the average scores and standard deviations ($\sigma$) of students on 18 items were used. Two tests were implemented to determine the differences between gender and education level in relation to students' affective learning outcomes. A Student's *t*-test for independent samples was used to calculate the statistical differences between gender and education level across the eight dimensions. Finally, our study also attempted to reveal whether the main factors of affective learning outcomes predict the academic learning outcomes in a blended format using linear regression analysis. We also compared the results of different groups of students, including male and female students, as well as postgraduate (PG) and undergraduate (UG) students.

## 3. Results

### 3.1. The Normality and Validity Testing

We analyzed the data normality test with the skewness and kurtosis values on each variable, which were between $-0.778$ to $-0.211$ and $-0.916$ to $5.794$, thus we concluded that the data observed was normally distributed. It has been established that the reliability test reflects the internal stability and consistent level of each measurement questionnaire. Therefore, a questionnaire with good reliability was obtained with a Cronbach $\alpha > 0.7$. In the current study, the Cronbach's $\alpha$ value on each construct was from 0.76–0.93. This confirmed the existence of high questionnaire reliability and internal consistency between latent variables. We also calculated the loading factor, C.R., and AVE to assess convergence validity. Convergent validity was indicated by an item factor loading $\geq 0.5$ [54], AVE $\geq 0.5$, and CR $\geq 0.7$ [55]. The AVE, loading factor, and C.R. values ranged from 0.62 to 0.91, from 0.72 to 0.96, and from 0.68 to 0.92, respectively, indicating very good convergent validity for this model (Table 2).

**Table 2.** Measurement model.

| Indicator | Items | Factor Loadings | $\alpha$ | C.R. | AVE |
|-----------|-------|-----------------|----------|------|-----|
| *PU* | PU1 | 0.804 | 0.895 | 0.910 | 0.816 |
| | PU2 | 0.821 | | | |
| | PU3 | 0.817 | | | |
| EU | EU1 | 0.879 | 0.901 | 0.892 | 0.837 |
| | EU2 | 0.883 | | | |
| PBC | PBC1 | 0.906 | 0.927 | 0.920 | 0.889 |
| | PBC2 | 0.904 | | | |
| | PBC3 | 0.911 | | | |
| *GLE* | GLE1 | 0.914 | 0.904 | 0.873 | 0.882 |
| | GLE2 | 0.876 | | | |

**Table 2.** *Cont.*

| Indicator | Items | Factor Loadings | α | C.R. | AVE |
|---|---|---|---|---|---|
| *KSI* | KSI1<br>KSI2 | 0.797<br>0.854 | 0.892 | 0.902 | 0.857 |
| *SSI* | SSI1<br>SSI2 | 0.811<br>0.847 | 0.872 | 0.819 | 0.797 |
| SP | SP1<br>SP2 | 0.724<br>0.801 | 0.760 | 0.682 | 0.624 |
| SAT | SAT1<br>SAT2 | 0.961<br>0.901 | 0.929 | 0.917 | 0.911 |
| AP | AP1<br>AP2<br>AP3<br>AP4 | 0.917<br>0.915<br>0.901<br>0.927 | 0.917 | 0.904 | 0.896 |

In addition, due to the cross-loading criterion, there is the presence of discriminant validity between all constructs as the loading indicators on their own constructs are in all cases higher than all their cross loadings with other constructs (Table 3).

**Table 3.** Cross-loading criterion.

| Constructs | PU | EU | PBC | GLE | KSI | SSI | SP | SAT | AP |
|---|---|---|---|---|---|---|---|---|---|
| **PU1** | **0.872** | 0.402 | 0.496 | 0.172 | 0.217 | 0.321 | 0.314 | 0.521 | 0.414 |
| **PU2** | **0.892** | 0.462 | 0.424 | 0.181 | 0.181 | 0.216 | 0.389 | 0.436 | 0.439 |
| **PU3** | **0.888** | 0.413 | 0.466 | 0.226 | 0.214 | 0.198 | 0.361 | 0.469 | 0.461 |
| **EU1** | 0.577 | **0.911** | 0.388 | 0.087 | 0.033 | 0.201 | 0.396 | 0.391 | 0.396 |
| **EU2** | 0.542 | **0.881** | 0.401 | 0.112 | 0.092 | 0.236 | 0.312 | 0.386 | 0.512 |
| **PBC1** | 0.436 | 0.534 | **0.865** | 0.236 | 0.155 | 0.134 | 0.276 | 0.168 | 0.476 |
| **PBC2** | 0.496 | 0.483 | **0.879** | 0.301 | 0.197 | 0.093 | 0.216 | 0.092 | 0.416 |
| **PBC3** | 0.479 | 0.514 | **0.894** | 0.258 | 0.167 | 0.110 | 0.389 | 0.179 | 0.389 |
| **GLE1** | 0.432 | 0.396 | 0.192 | **0.792** | 0.409 | 0.373 | 0.413 | 0.093 | 0.401 |
| **GLE2** | 0.422 | 0.414 | 0.206 | **0.844** | 0.392 | 0.416 | 0.373 | 0.116 | 0.373 |
| **KSI1** | 0.408 | 0.363 | 0.311 | 0.411 | **0.816** | 0.288 | 0.385 | 0.189 | 0.341 |
| **KSI2** | 0.369 | 0.355 | 0.295 | 0.401 | **0.895** | 0.233 | 0.403 | 0.204 | 0.333 |
| **SSI1** | 0.417 | 0.351 | 0.418 | 0.379 | 0.366 | **0.896** | 0.385 | 0.116 | 0.285 |
| **SSI2** | 0.324 | 0.399 | 0.374 | 0.219 | 0.492 | **0.841** | 0.326 | 0.241 | 0.217 |
| **SP1** | −0.012 | 0.127 | 0.085 | 0.205 | 0.544 | 0.363 | **0.815** | 0.278 | 0.119 |
| **SP2** | −0.019 | 0.092 | 0.032 | 0.116 | 0.487 | 0.431 | **0.883** | 0.211 | 0.183 |
| **SAT1** | 0.379 | 0.335 | 0.201 | 0.082 | 0.118 | 0.264 | 0.179 | **0.916** | 0.179 |
| **SAT2** | 0.511 | 0.391 | 0.188 | 0.113 | 0.074 | 0.312 | 0.201 | **0.895** | 0.201 |
| **AP1** | 0.192 | 0.169 | 0.176 | 0.148 | 0.239 | 0.246 | 0.139 | 0.479 | **0.839** |
| **AP2** | 0.211 | 0.197 | 0.206 | 0.110 | 0.138 | 0.174 | 0.168 | 0.314 | **0.878** |
| **AP3** | 0.279 | 0.113 | 0.093 | 0.086 | 0.236 | 0.116 | 0.191 | 0.416 | **0.891** |
| **AP4** | 0.186 | 0.176 | 0.119 | 0.129 | 0.265 | 0.162 | 0.104 | 0.362 | **0.904** |

### 3.2. Students' Affective Learning Outcomes and Academic Performance for Different Genders

We used an independent sample *t*-test to determine the differences between male and female students' affective learning outcomes and academic performance in a blended environment. Table 4 summarizes the results.

According to Table 4, the female students' mean scores were generally higher than the male students' mean scores for almost all considered indicators of affective learning outcomes and academic performance in a blended learning environment. The differences between the two tested groups were significant in two indicators—ease of use ($p < 0.05$) and academic performance ($p < 0.01$). Males experienced blended learning with less difficulty, while the average academic performance scale was higher for females.

**Table 4.** Gender differences in students' affective learning outcomes and academic performance.

| Factors | | | Gender | | *t*-Value | *p*-Value |
|---|---|---|---|---|---|---|
| | | | Male (SD) | Female (SD) | | |
| Affective learning outcomes | Perceived experience (PE) | PU | 3.88 (0.65) | 3.93 (0.69) | −1.11 | 0.264 |
| | | EU | 3.77 (0.66) | 3.69 (0.73) | 2.59 * | 0.042 |
| | | PBC | 3.55 (0.72) | 3.49 (0.67) | 1.52 | 0.184 |
| | Perceived benefit (PB) | GLE | 3.75 (0.71) | 3.80 (0.69) | −1.45 | 0.174 |
| | | KSI | 3.44 (0.68) | 3.46 (0.70) | −0.98 | 0.842 |
| | | SSI | 3.55 (0.65) | 3.60 (0.67) | −1.51 | 0.189 |
| | | SP | 3.70 (0.69) | 3.73 (0.67) | −1.14 | 0.781 |
| | Satisfaction (SAT) | | 3.91 (0.73) | 3.87 (0.69) | 1.37 | 0.211 |
| Academic performance (AP) | | | 3.41 (0.75) | 3.64 (0.67) | −6.76 ** | 0.007 |

* $p < 0.05$; ** $p < 0.01$.

### 3.3. Students' Affective Learning Outcomes and Academic Performance for Different Education Level

We also used an independent sample *t*-test to investigate the differences between undergraduates' and postgraduates' affective learning outcomes and academic performance in a blended environment. Table 5 presents the results.

**Table 5.** The education level differences in students' affective learning outcomes and academic performance.

| Factors | | | Education Level | | *t*-Value | *p*-Value |
|---|---|---|---|---|---|---|
| | | | UG (SD) | PG (SD) | | |
| Affective learning outcomes | Perceived experience (PE) | PU | 3.89 (0.70) | 4.11 (0.68) | −6.65 ** | 0.007 |
| | | EU | 3.73 (0.65) | 3.76 (0.60) | −0.37 | 0.747 |
| | | PBC | 3.44 (0.73) | 3.53 (0.76) | −2.73 * | 0.037 |
| | Perceived benefit (PB) | GLE | 3.44 (0.73) | 3.53 (0.76) | −2.73 * | 0.037 |
| | | KSI | 3.45 (0.65) | 3.40 (0.67) | 1.96 * | 0.048 |
| | | SSI | 3.57 (0.71) | 3.62 (0.70) | −1.98 | 0.046 |
| | | SP | 3.70 (0.66) | 3.81 (0.72) | −3.46 * | 0.018 |
| | Satisfaction (SAT) | | 3.80 (0.68) | 3.94 (0.65) | −4.79 ** | 0.009 |
| Academic performance (AP) | | | 3.48 (0.75) | 3.56 (0.70) | −2.63 * | 0.041 |

* $p < 0.05$; ** $p < 0.01$.

The positive value of the mean difference demonstrates that the mean scores of the undergraduates are higher than those of the postgraduates, while the negative value reflects the reverse. According to the results, UG and PG students showed significant differences in almost all measurement factors, including perceived usefulness ($p < 0.01$), perceived behavioral control ($p < 0.05$), general learning effectiveness ($p < 0.05$), study skills improvement ($p < 0.05$), sense of progress ($p < 0.05$), satisfaction ($p < 0.01$), and academic performance ($p < 0.05$).

### 3.4. Regression Analysis

The results of the linear regression analysis conducted to analyze whether the affective learning outcomes of the university students predicted the academic performance scales are presented in Table 6.

**Table 6.** Affective leaning outcomes as a predictor of academic outcomes for male and female students.

| | **Males (N = 241)** | | | | | | |
|---|---|---|---|---|---|---|---|
| | **B** | **SEB** | **β** | **_t_** | **F** | **R$^2$** | **Adjusted R$^2$** |
| Constant | 2.77 | 0.09 | | 6.02 ** | 30.29 ** | 0.38 | 0.35 |
| PE | 0.01 | 0.00 | 0.29 | 5.89 ** | | | |
| | **B** | **SEB** | **β** | **_t_** | **F** | **R$^2$** | **Adjusted R$^2$** |
| Constant | 3.17 | 0.10 | | 7.09 ** | 33.95 ** | 0.31 | 0.30 |
| PB | 0.01 | 0.00 | 0.31 | 6.47 ** | | | |
| | **B** | **SEB** | **β** | **_t_** | **F** | **R$^2$** | **Adjusted R$^2$** |
| Constant | 4.02 | 0.12 | | 11.34 *** | 57.29 *** | 0.58 | 0.52 |
| SAT | 0.01 | 0.00 | 0.43 | 10.18 *** | | | |
| | **Females (N = 272)** | | | | | | |
| | **B** | **SEB** | **β** | **_t_** | **F** | **R$^2$** | **Adjusted R$^2$** |
| Constant | 1.96 | 0.08 | | 4.58 * | 20.07 * | 0.19 | 0.17 |
| PE | 0.01 | 0.00 | 0.23 | 3.37 * | | | |
| | **B** | **SEB** | **β** | **_t_** | **F** | **R$^2$** | **Adjusted R$^2$** |
| Constant | 3.61 | 0.10 | | 7.81 ** | 37.53 ** | 0.36 | 0.31 |
| PB | 0.01 | 0.00 | 0.34 | 6.14 ** | | | |
| | **B** | **SEB** | **β** | **_t_** | **F** | **R$^2$** | **Adjusted R$^2$** |
| Constant | 4.42 | 0.13 | | 12.87 *** | 60.29 *** | 0.63 | 0.60 |
| SAT | 0.01 | 0.00 | 0.44 | 11.56 *** | | | |

Dependent variable: Academic performance. Note: * $p < 0.05$; ** $p < 0.01$; *** $p < 0.001$.

According to Table 6, the affective learning outcomes predict the academic performance results of the students in a significantly positive way (β ranged from 0.23 to 0.44; t ranged from 4.58 to 12.87). The results between males and females are similar and the academic performance results were mostly predicted by satisfaction level (R$^2$ = 0.58, adjusted R$^2$ = 0.52 for males, R$^2$ = 0.63, adjusted R$^2$ = 0.60 for females).

The results of the linear regression analyses conducted to analyze whether the affective learning outcomes of the university undergraduate and graduate students predicted the academic performance scales are presented in Table 7.

Table 7 summarizes the affective learning outcomes predicting the academic performance of the undergraduate and postgraduate students in a positive way (β ranged from 0.22 to 0.46, and t ranged from 4.51 to 13.27). A difference was revealed between undergraduate students' results and postgraduate students' results. A total of 44% of the total variance of the academic performance scores for postgraduate students can be explained by the perceived benefit scores (R$^2$ = 0.59, adjusted R$^2$ = 0.55), while only 25% of the total variance of the academic performance scores for undergraduate students can be explained by the perceived benefit scores (R$^2$ = 0.23, adjusted R$^2$ = 0.20). Thus, perceived benefit has a greater effect on academic performance for postgraduate students. A total of 46% of the total variance of the academic performance scores for undergraduate students can be explained by the satisfaction level (R$^2$ = 0.60, adjusted R$^2$ = 0.57), while only 32% of the total variance of the academic performance scores for postgraduate students can be explained by the satisfaction level (R$^2$ = 0.37, adjusted R$^2$ = 0.34). Thus, satisfaction level has a greater effect on academic performance results for undergraduate students than for postgraduate students.

Generally, it was found that satisfaction was the factor that mostly influenced students' academic performance in a blended environment in this research.

**Table 7.** Affective learning outcomes as a predictor of academic performance for undergraduate and graduate students.

| | B | SEB | β | t | F | $R^2$ | Adjusted $R^2$ |
|---|---|---|---|---|---|---|---|
| **Undergraduates (N = 403)** | | | | | | | |
| | **B** | **SEB** | **β** | **t** | **F** | **$R^2$** | **Adjusted $R^2$** |
| Constant | 2.55 | 0.10 | | 5.94 ** | 30.11 ** | 0.38 | 0.35 |
| PE | 0.01 | 0.00 | 0.28 | 5.77 ** | | | |
| | **B** | **SEB** | **β** | **t** | **F** | **$R^2$** | **Adjusted $R^2$** |
| Constant | 2.31 | 0.09 | | 4.73 * | 21.29 ** | 0.23 | 0.20 |
| PB | 0.01 | 0.00 | 0.25 | 5.91 * | | | |
| | **B** | **SEB** | **β** | **t** | **F** | **$R^2$** | **Adjusted $R^2$** |
| Constant | 4.63 | 0.13 | | 13.27 *** | 61.29 *** | 0.60 | 0.57 |
| SAT | 0.01 | 0.00 | 0.46 | 12.49 *** | | | |
| **Postgraduates (N = 110)** | | | | | | | |
| | **B** | **SEB** | **β** | **t** | **F** | **$R^2$** | **Adjusted $R^2$** |
| Constant | 2.18 | 0.08 | | 4.51 * | 20.07 * | 0.19 | 0.17 |
| PE | 0.01 | 0.00 | 0.22 | 4.18 * | | | |
| | **B** | **SEB** | **β** | **t** | **F** | **$R^2$** | **Adjusted $R^2$** |
| Constant | 4.98 | 0.11 | | 11.09 *** | 58.31 *** | 0.59 | 0.55 |
| PB | 0.01 | 0.00 | 0.44 | 9.93 *** | | | |
| | **B** | **SEB** | **β** | **t** | **F** | **$R^2$** | **Adjusted $R^2$** |
| Constant | 3.91 | 0.10 | | 7.66 ** | 40.18 ** | 0.37 | 0.34 |
| SAT | 0.01 | 0.00 | 0.32 | 6.04 ** | | | |

Dependent variable: Academic performance. Note: * $p < 0.05$; ** $p < 0.01$; *** $p < 0.001$.

## 4. Discussion

This study aimed to examine students' affective learning outcomes and academic performance in a blended learning environment. Nine measurement items were used for the study to analyze gender differences and differences between undergraduate and postgraduate students within our research model. We also investigated whether gender or education level significantly influences the relationship between affective learning outcomes and academic performance.

According to the results obtained, female students, on average, showed higher scores than male students in all considered indicators of affective learning outcomes and academic achievement in blended learning settings. Differences between the two test groups were significant in two dimensions—ease of use and academic performance. Men experienced less difficulty with blended learning, while the average academic achievement scale was higher for women. It can be concluded that men are more receptive and adaptive to the blended environment. However, this did not allow them to get higher results in academic achievements. Female students, despite more difficulties with blended learning, were able to learn the material better.

The study also showed that postgraduate students perceive blended learning much better than undergraduates. Postgraduate students showed significant differences in almost all measured factors, including perceived usefulness, perceived behavioral control, general learning effectiveness, knowledge sharing and increasing, sense of progress, satisfaction, and academic achievement. This can be explained by the fact that postgraduate students have gone through a longer educational path, have more experience in obtaining knowledge, which helps them learn more effectively in a mixed environment. Thus, it is worth paying more attention to undergraduates when including them in blended learning.

Based on the study, it can be concluded that, as in previous studies [25,26,32,49], the selected factors do have a significant impact on affective learning outcomes in a blended environment. The analysis agrees with the findings from other studies [34,37,38]. A dis-

tinguishing feature of the current study is the use of moderating factors such as gender and education level. It is important to note that educational technologies do not always work equally effectively for all groups of students. Past research [13,16,19,28] has identified scoring factors that are truly relevant to current research and have a significant impact on academic achievement and affective learning outcomes. Thanks to previous studies [7,8,35,48], we were able to form a model for the current study. However, the contribution of current work in the study is in the categories of students.

However, a distinctive feature of this study is the study of the significance of the considered factors on the results of students in male and female groups separately, as well as separately for undergraduate and postgraduate students. Thus, the practical application of the results of this study will be to the category of students in the implementation of blended learning. So, we know that it is more difficult for a group of female students to study in a blended environment, however, they cope no worse than a group of male students, but perhaps for female students the learning process becomes more labor-intensive. Undergraduate students also need to be treated more carefully, offering them possible additional instructions and training materials, as well as more sensitive supervision.

## 5. Conclusions

The findings of the current study have several important implications for educators when implementing live online learning in the future. It can be used by university leaders and teachers, who will be able to direct all their efforts to improving students' satisfaction with blended learning, since satisfaction has a direct impact on the affective learning outcomes in a blended environment. Based on the results of the study, educational program developers can structure courses by discipline in such a way that a mixed environment has a positive effect on affective learning outcomes and student satisfaction. An important task is to organize blended learning in such a way that students (especially undergraduate students) feel the benefits of taking the course, are motivated to receive and share knowledge, feel progress, and increase their own knowledge base. To do this, it is important to distribute the course and balance between the online environment and face-to-face classes so that students feel the effectiveness of learning.

The limitations of the study are associated with the sample, as we investigated students' perceptions and behaviors only in Russia while the COVID-19 pandemic and blended learning are common worldwide. The study was based on humanities students, while students from technical institutions could have their own characteristics, and their affective learning outcomes and academic performance results in a blended environment may differ. It is also important to note that for such an analysis, it is possible to use other approaches related to mediation and moderation effects. This study was based only on differences in education and gender, but there are other characteristics that require special attention.

**Author Contributions:** Conceptualization, A.K. and E.T.; methodology, A.K.; software, T.B.; validation, A.K., E.T. and T.B.; formal analysis, A.K.; investigation, E.T.; resources, T.B.; data curation, T.B.; writing—original draft preparation, A.K.; writing—review and editing, A.K.; visualization, E.T.; supervision, T.B.; project administration, A.K.; funding acquisition, E.T. All authors have read and agreed to the published version of the manuscript.

**Funding:** The research is partially funded by the Ministry of Science and Higher Education of the Russian Federation under the strategic academic leadership program "Priority 2030" (Agreement 075-15-2021-1333 dated 30 September 2021).

**Institutional Review Board Statement:** The study was conducted in accordance with the Declaration of Helsinki, and approved by the Institutional Review Board of Institute of Humanities (protocol code 715, 1 September 2021).

**Informed Consent Statement:** Informed consent was obtained from all subjects involved in the study.

**Data Availability Statement:** Not applicable.

**Conflicts of Interest:** The authors declare no conflict of interest.

## Appendix A

Students measured "To what extent do you agree or disagree with the following statements".

| № | Construct | Scale |
|---|-----------|-------|
| 1 | Perceived usefulness | PU 1 I believe that using blended learning technologies would improve my ability to learn<br>PU 2 I believe that blended learning technologies would allow me to get my work done more quickly<br>PU 3 I believe that blended format would be useful for my learning |
| 2 | Ease of use | EU 1 It was easy for me to learn in blended environment<br>EU 2 I do not notice any inconsistencies as I learn in blended environment |
| 3 | Perceived behavioral control | PBC 1 I have sufficient extent of knowledge to use blended learning<br>PBC 2 I have sufficient extent of control to make a decision to adopt blended learning<br>PBC 3 I have sufficient extent of self-confidence to make a decision to adopt blended learning |
| 4 | General learning effectiveness | GLE 1 I achieved the objectives of the learning program in a blended environment<br>GLE 2 The quality of the learning course in blended environment was high. |
| 5 | Knowledge sharing and increasing | KSI 1 My overall professional knowledge increased after the blended learning course<br>KSI 2 I was able to share my knowledge with peers during the blended learning course |
| 6 | Study skill improvement | SSI 1 Due to the blended learning course I improved my time management skills<br>SSI 2 I Due to the blended learning course I improved my problem-solving skills |
| 7 | Sense of progress | SP 1 I feel the general improvement in my knowledge and skills after the blended learning course<br>SP 2 I feel a progress of my professional development after the blended learning course |
| 8 | Satisfaction | SAT 1 My overall experience with blended learning was very satisfying<br>SAT 2 My overall experience with blended learning was very pleasing |

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
