# Peer review of "Students’ Affective Learning Outcomes and Academic Performance in the Blended Environment at University: Comparative Study"

_sustainability, doi:10.3390/su141811341_

Round 1

Reviewer 1 Report

Please follow the comments in the attached file. I hope that the answers to these remarks will lead to a better understanding of your work.

Reviewer 2 Report

Congratulation on the articles idea, its realization and the added value it brings!

Maybe it would be a plus for the study if:

- in subchapter 1.1.2 would be paid attention to the analysis of the elements of the perceived experience and perceived benefit  as they appear in Figure 1;

- the would be more details on how the questionnaire was applied online and the data were collected;

- it is possible, based on the obtained data, to extend some conclusions with implications on the improvement of academic performances from the perspective of affective learning outcomes.

Reviewer 3 Report

Thank you for allowing me to review your paper. Please find the feedback that I believe would help you to improve the manuscript.

1.     The introduction section needs to be re-written. It is not clear from the introduction why this study is being conducted. Please mention the gap that the study is addressing. Please write why you are conducting this study, and cite the existing studies conducted on blended learning. Also, try to showcase how your study is different from those and what your contribution is to the body of literature. Then introduce the research questions.

2.     The section written as the theoretical framework is the Background, so please change the heading to Background. Please mention why affective outcomes are important in a blended setting. What has been said in other studies about affective outcomes in the context of blended learning? 

3.     A section describing the learning design will help readers to understand the blended learning context.

4.     Result section: My suggestion is to incorporate the complete diagram with each construct and items and their corresponding factor loadings so that one can clearly note the factor loading for each item. Please create a table for cross-loading as well. As of now, from the paper, it is not clear if there was a cross-loading for the items. If there were any cross-loadings, do mention how did you handle them and how that impacted the model.

5.     Please create a table with each item of the survey demonstrating their AVE, factor loading, and CR values. Also, cite the literature or other studies that mention the acceptable range for AVE, factor loading, and CR values.

6.     The discussion section needs to be re-written and cite more studies and discuss your findings. Please discuss and describe how your findings are different or similar to other studies.

7.     I would suggest adding a conclusion section to highlight the take-home message of the study.

Round 2

Reviewer 1 Report

The author provides an improved way to resubmit that addresses my previous concerns. The text has been organized and restructured. It has greater coherence which facilitates the understanding of the reader.

The following are the places that need to be modified in this article.

There are 3 satisfaction questions in Appendix A, but only SAT1 and SAT2 in Tables 2 and 3.

Reviewer 3 Report

Dear Authors,

Thank you for making the changes. The changes look good. 

Also, I would suggest adding citations to the introduction section.

Thanks!
